# The *Xenopus* Oocyte System: Molecular Dynamics of Maturation, Fertilization, and Post-Ovulatory Fate

**DOI:** 10.3390/biom16010022

**Published:** 2025-12-23

**Authors:** Ken-Ichi Sato

**Affiliations:** Laboratory of Oocyte Biology, Faculty of Life Sciences, Kyoto Sangyo University, Kamigamo-Motoyama, Kita-ku, Kyoto 603-8555, Japan; kksato@cc.kyoto-su.ac.jp

**Keywords:** *Xenopus* oocyte, MPF (maturation-promoting factor), MAPK signaling, calcium wave, cytoplasmic determinants, fertilization and egg activation, oocyte atresia, developmental competence

## Abstract

The *Xenopus* oocyte has long served as a versatile and powerful model for dissecting the molecular underpinnings of reproductive and developmental processes. Its large size, manipulability, and well-characterized cell cycle states have enabled generations of researchers to illuminate key aspects of oocyte maturation, fertilization, and early embryogenesis. This review provides an integrated overview of the cellular and molecular events that define the *Xenopus* oocyte’s transition from meiotic arrest to embryonic activation—or alternatively, to programmed demise if fertilization fails. We begin by exploring the architectural and biochemical landscape of the oocyte, including polarity, cytoskeletal organization, and nuclear dynamics. The regulatory networks governing meiotic resumption are then examined, with a focus on MPF (Cdk1/Cyclin B), MAPK cascades, and translational control via CPEB-mediated cytoplasmic polyadenylation. Fertilization is highlighted as a calcium-dependent trigger for oocyte activation. During fertilization in vertebrates, sperm-delivered phospholipase C zeta (PLCζ) is a key activator of Ca^2+^ signaling in mammals. In contrast, amphibian species such as *Xenopus* lack a PLCZ1 ortholog and instead appear to rely on alternative protease-mediated signaling mechanisms, including the uroplakin III–Src tyrosine kinase pathway and matrix metalloproteinase (MMP)-2 activity, to achieve egg activation. The review also addresses the molecular fate of unfertilized eggs, comparing apoptotic and necrotic mechanisms and their relevance to reproductive health. Finally, we discuss recent innovations in *Xenopus*-based technologies such as mRNA microinjection, genome editing, and in vitro ovulation systems, which are opening new avenues in developmental biology and translational medicine. By integrating classic findings with emerging frontiers, this review underscores the continued value of the *Xenopus* model in elucidating the fundamental processes of life’s origin. We conclude with perspectives on unresolved questions and future directions in oocyte and early embryonic research.

## 1. Introduction

The *Xenopus* frog, particularly *Xenopus laevis*, has long served as a cornerstone of developmental and cell biological research [1,2]. Its large, manipulable oocytes, synchronous embryogenesis, and ease of in vitro handling have rendered it a widely used and versatile system for elucidating the molecular events underpinning fertilization, early development, and cellular signaling [3,4]. In recent decades, *Xenopus* research has expanded from classical embryological inquiries to encompass molecular genetics, cell cycle regulation, and translational medicine, thereby highlighting its continued relevance in the post-genomic era and in regenerative research [5,6,7].

At the heart of this model system lies the oocyte—an exceptional cell that encapsulates not only the maternal genome but also a repertoire of localized mRNAs, proteins, and organelles that prefigure the embryo’s developmental competence [8,9]. From the arrest of meiosis in prophase I to the cascade of signaling events triggered by progesterone-induced maturation, the *Xenopus* oocyte provides a powerful system to study cell cycle transitions, cytoplasmic organization, and the dynamic interplay between cell fate determinants and cytoskeletal architectures [10,11,12,13,14,15].

Fertilization in *Xenopus* further unveils a tightly choreographed biochemical cascade. The entry of a single spermatozoon triggers a global calcium wave across the oocyte cortex [16,17], launching a series of downstream events including cortical granule exocytosis, the establishment of polyspermy blocks, and the re-entry into the embryonic cell cycle [11,18,19,20,21]. This fertilization-induced activation is not merely a binary switch but a multilayered program involving the inactivation of cytostatic factors (CSF), stabilization of maternal mRNAs, and spatially resolved activation of translation pathways [22,23,24,25].

The transition from a fertilized egg to a competent zygote also involves the utilization of maternal patterning determinants. Vegetally localized mRNAs such as VegT and Vg1 [26,27], together with cortical rotation-induced nuclear translocation of β-catenin [28,29], underpin axis formation and germ layer specification. These determinants, long-studied using microinjection and lineage tracing in *Xenopus* [30,31], are now being reinvestigated using modern approaches including CRISPR/Cas9-mediated genome editing and RNA-seq-based transcriptomics [6,32,33].

However, not all oocytes reach their developmental competence. The fate of unfertilized eggs, often overlooked in developmental discourse, offers a unique opportunity to investigate programmed cell death, mitochondrial integrity, and oocyte lifespan regulation [34]. Apoptosis and other forms of cell demise, including overactivation-induced necrosis-like pathways, are essential to understand the fine balance between oocyte survival and quality control [35,36], an area of increasing significance in reproductive medicine [37,38].

This review integrates classical and contemporary studies on *Xenopus* oocytes and eggs, providing a continuum from meiotic arrest and maturation to fertilization, early development, and programmed death [39,40,41,42]. Each section emphasizes not only the biological principles but also the experimental paradigms that have shaped our current understanding [2,3]. Special attention is given to the temporal and spatial regulation of key signaling cascades such as the MPF-Mos-mitogen-activated protein kinase (MAPK) axis [20,25], the role of maternal mRNAs and cytoskeletal dynamics [9,43], and the mechanistic basis of fertilization-triggered cell cycle resumption [22,44].

By focusing on the *Xenopus* oocyte as a biological and methodological interface, this review aims to provide a unified perspective that spans developmental cell biology, molecular signaling, and biomedical applications [5,7]. In doing so, it seeks to inspire future research that builds upon and contributes to the rich legacy of *Xenopus* as a model organism [6].

## 2. The Power and Appeal of *Xenopus* Oocytes as a Model System

### 2.1. Why Xenopus laevis Has Been a Cornerstone in Developmental and Cell Biology

The African clawed frog, *Xenopus laevis*, has long served as a cornerstone model in developmental and cell biology [1,5]. Its prominence is owed to several biological and practical advantages. Most notably, the species produces large, externally fertilized eggs—approximately 1.2 mm in diameter—that facilitate microinjection and manipulation with relative ease [2,3]. These oocytes are optically accessible and can be maintained in vitro under controlled conditions, allowing for dynamic observation of physiological and developmental processes [4].

The accessibility of *Xenopus* oocytes in large quantities, owing to their year-round availability and ease of hormonal induction, makes them particularly well-suited for biochemical assays and molecular screenings [12]. Their evolutionary proximity to mammals, while retaining a simpler embryonic structure, offers a unique vantage point to interrogate conserved signaling pathways and cellular mechanisms [7].

Historically, *Xenopus* has provided fundamental insights into nuclear reprogramming, signal transduction, and cell cycle regulation [14,45]. Despite the emergence of other model organisms such as zebrafish and mouse [46,47], *Xenopus* continues to offer distinctive advantages in early developmental research. More recently, *Xenopus tropicalis*, possessing a diploid genome and enhanced amenability to genetic manipulation, has expanded the experimental toolkit, particularly with the advent of genome editing technologies [32,48,49].

### 2.2. Cytological Architecture of Xenopus Oocytes

A defining feature of *Xenopus* oocytes is their pronounced polarity and highly organized cytoplasm [31,50,51]. The animal and vegetal hemispheres are morphologically and functionally distinct. The animal pole, characterized by a darker pigmentation due to melanin granules, houses the germinal vesicle (nucleus) and is enriched in ribosomes and mitochondria [4,50]. In contrast, the vegetal pole is paler and contains abundant yolk platelets that serve as nutrient stores for embryonic development [52,53].

The cytoskeleton is central to maintaining this polarity, with microtubules organized radially from the microtubule organizing center, actin filaments supporting cortical integrity and vesicular transport, and intermediate filaments contributing to mechanical stability [54,55]. The polarized distribution of organelles and maternal mRNAs—such as VegT and Vg1—provides critical determinants for axis establishment and germ layer specification even before fertilization occurs [26,27,56].

Another unique attribute is the pigment granule distribution, which allows researchers to readily orient and manipulate oocytes during experimental procedures [57]. Overall, the cytoplasmic organization of *Xenopus* oocytes provides an ideal platform for investigating the spatial regulation of signal transduction and mRNA translation [9].

### 2.3. Overview of Research Techniques in Xenopus Oocytes

The versatility of *Xenopus* oocytes as a model system is amplified by a wide range of sophisticated experimental techniques. Microinjection remains a cornerstone methodology, allowing the introduction of RNA, DNA, morpholinos, or proteins into specific regions of the oocyte to assess function or induce phenotypes [2,58]. The large size of the oocyte facilitates fine control over injection volume and localization, enabling region-specific perturbation studies.

Nuclear transfer techniques, pioneered in part through *Xenopus*, have also enabled major advances in understanding nuclear reprogramming and epigenetic regulation. Somatic cell nuclei transplanted into enucleated oocytes can reinitiate developmental programs, providing insights into cellular plasticity and differentiation [45,59].

Another key experimental platform is the in vitro ovulation and maturation system. Hormonal cues, particularly progesterone, can trigger maturation from the germinal vesicle stage to metaphase II, thereby recapitulating the transition toward fertilization [12,14]. This system permits detailed analysis of signal transduction pathways, such as MPF activation and MAPK signaling, in a temporally controlled manner [20,25,60]. Building upon this, recent studies have established an in vitro ovulation reconstruction model, in which progesterone treatment combined with low concentrations of MMPs synchronously induces both germinal vesicle breakdown (GVBD) and follicle rupture [61]. This experimental advance demonstrated that MAPK activity is essential for both GVBD and follicular rupture, while MMP activity specifically mediates follicular rupture, thereby highlighting the coordinated but distinct contributions of these pathways. Subsequent reviews have emphasized how such ex vivo and in vitro approaches faithfully reproduce ovulation across species, situating *Xenopus* as a valuable comparative model for dissecting the universal molecular and cytological events of ovulation [62].

Electrophysiological assays, especially voltage-clamp recordings, are also frequently applied to *Xenopus* oocytes for the functional characterization of ion channels and membrane proteins [63,64,65]. Together, these methodological strengths—including the emerging ovulation reconstruction system—solidify *Xenopus* as an indispensable model for integrative cell biology.

### 2.4. Limitations and Expanding Applications of Xenopus as a Model Organism—A Comparative Perspective with Xenopus tropicalis

Despite its many advantages, *Xenopus laevis* is not without limitations as a model organism. One of the primary concerns is its allotetraploid genome, which complicates genetic analyses and particularly hampers straightforward loss-of-function approaches [6]. The complexity of redundant gene copies remains a critical barrier in the genomics era of developmental biology (Table 1).

In contrast, *Xenopus tropicalis*, a closely related species, offers a diploid genome that is more amenable to precise genetic manipulation [49,66]. The shorter generation time and smaller body size of *X. tropicalis* make it well-suited for forward and reverse genetic approaches, including CRISPR/Cas9-mediated gene editing [32]. These features have increasingly positioned *X. tropicalis* as a complementary system for investigations requiring genetic tractability (Table 1).

Nevertheless, *X. laevis* retains supremacy in several domains. Its larger oocytes remain the gold standard for biochemical assays, electrophysiology, and high-volume injection-based experiments [63,64,65]. Furthermore, the well-characterized oocyte maturation and fertilization system in *X. laevis* provides a unique platform for dissecting meiotic cell cycle transitions and signal transduction pathways [12,14].

Comparative studies that exploit the complementary strengths of both species—*X. tropicalis* for high-throughput genetic screens and *X. laevis* for mechanistic dissection—have emerged as an effective dual-species strategy [7]. As emerging technologies such as single-cell transcriptomics and proteomics are adapted for amphibian models, integrating *X. laevis* and *X. tropicalis* promises increasingly sophisticated analyses of developmental and reproductive biology [67,68] (Table 1).

**Table 1 biomolecules-16-00022-t001:** Comparative features of *Xenopus laevis* and *Xenopus tropicalis*.

Feature	*Xenopus laevis*	*Xenopus tropicalis*	Reference(s)
Genome ploidy	Allotetraploid (2n = 36 × 2)	Diploid (2n = 20)	[6,66]
Genome size	~3.1 Gb	~1.7 Gb	[6,66]
Generation time	~1 year	~4–6 months	[5]
Embryonic development rate (23 °C in *Xenopus laevis*/26 °C in *Xenopus tropicalis*)	~24 h to neurula	~18 h to neurula	[2,51]
Oocyte diameter (Stage VI)	~1.2 mm	~0.7 mm	[7,50]
Ease of oocyte microinjection	Excellent (large oocytes)	Good (smaller oocytes)	[50,64]
Suitability for transgenesis/CRISPR	Limited by tetraploidy	High; efficient germline transmission	[32,69,70]
Genome resources and annotation	Well-established ESTs and partial genome	Fully sequenced, annotated genome	[66]
Use in developmental biology	Classic model for cell cycle, embryology, and oocyte maturation	Complementary model for forward genetics and genome editing	[3,5]
Main advantages	Large eggs, historical knowledge base, ease of manipulation	Genetic tractability, diploid genome, shorter generation time	[5]
Main limitations	Polyploid genome complicates genetics	Smaller size, limited historical datasets	[5,7]

The table summarizes key biological and experimental characteristics of the two *Xenopus* species, emphasizing their complementary strengths for developmental and molecular studies.

### 2.5. Summary

*Xenopus* laevis oocytes have long served as a foundational model in developmental and cell biology due to their unique combination of size, manipulability, and experimental versatility. This chapter introduces the biological and practical features that make *Xenopus* oocytes particularly valuable. The large size (approximately 1 mm in diameter) allows for direct visualization and manipulation of subcellular components without advanced microscopy, while their accessibility and abundance make them an ideal platform for large-scale biochemical assays and microinjection-based experiments.

The chapter also delineates the oocyte’s polarized structure, marked by a distinct animal-vegetal axis, cytoplasmic localization of maternal determinants, and pigment patterning. These characteristics are critical for understanding early embryonic axis formation and the spatial regulation of developmental signals.

In addition to their biological utility, the chapter outlines commonly used research techniques such as nuclear transplantation, in vitro maturation systems, and mRNA microinjection protocols. These have enabled breakthroughs in understanding cell cycle regulation, signaling cascades, and translational control.

Finally, the chapter discusses limitations of the *Xenopus* laevis model—such as tetraploidy and long generation times—and compares it with *Xenopus tropicalis*, a diploid relative with a sequenced genome and improved genetic tractability. This provides a balanced view of how *Xenopus* species can be strategically utilized in contemporary research frameworks.

## 3. The Regulation of Oocyte Formation and Meiotic Arrest

### 3.1. Developmental Stages of Xenopus Oocytes and Meiotic Arrest at the GV Stage

In the ovary of *Xenopus laevis*, oocytes progress through six morphologically distinct stages, as originally described by Dumont [50]. These stages range from stage I, a small translucent oocyte, to stage VI, a fully grown oocyte ready for maturation. Importantly, these oocytes arrest at the prophase of meiosis I, characterized by the presence of a large nucleus known as the germinal vesicle (GV) [14]. This prolonged arrest can last for months and is tightly regulated by both intrinsic and extrinsic factors.

During this stage, the oocyte remains transcriptionally active and accumulates maternal mRNAs, proteins, and organelles necessary for subsequent embryonic development [53]. The arrest is maintained by a low activity state of MPF, composed of Cyclin B and Cdk1, and is supported by the action of regulatory kinases such as Myt1, together with inhibitory phosphorylations on Cdk1 [71,72].

This arrest provides a valuable window for experimental manipulation and observation, enabling researchers to inject RNAs, proteins, or other factors and monitor their effects on cellular physiology without the complication of cell division [73,74].

The germinal vesicle itself is a prominent and accessible structure that can be directly visualized and manipulated, which facilitates a wide array of experimental designs, including nuclear transplantation and chromatin remodeling studies [45,75]. Overall, the GV-stage oocyte is a uniquely versatile tool in the arsenal of developmental biology research (Figure 1A).

### 3.2. The Role of MPF (Cdk1/CyclinB Complex)

The resumption of meiosis in *Xenopus* oocytes is governed by a key molecular engine known as MPF, which comprises the cyclin-dependent kinase Cdk1 (also known as p34^cdc2^) and its regulatory partner Cyclin B [71,76,77]. MPF plays a central role in initiating GVBD, spindle formation, and the progression to metaphase II, where the oocyte remains arrested until fertilization [78,79].

In fully grown GV-stage oocytes, MPF remains inactive due to inhibitory phosphorylation of Cdk1 by kinases such as Wee1 and Myt1 [72,80]. The hormonal cue—typically progesterone—triggers a signaling cascade leading to the activation of Cdc25 phosphatase, which removes the inhibitory phosphate groups, thereby activating Cdk1 [81]. Concurrently, Cyclin B synthesis is upregulated and protected from degradation, further contributing to MPF activity [71,82].

A notable feature of MPF regulation is its bistability and positive feedback: once activated, MPF enhances Cdc25 activity and inhibits Wee1/Myt1, creating a switch-like behavior essential for the rapid and irreversible progression of meiosis [83,84]. This switch ensures that oocyte maturation proceeds decisively and synchronously across a population of cells.

Experimental identification of MPF activity was historically confirmed through cytoplasmic transfer experiments. Injection of cytoplasm from matured oocytes into immature ones could induce GVBD, demonstrating the presence of a transferable maturation factor [78,79]. More refined biochemical assays, such as the histone H1 kinase assay, have been employed to quantify MPF activity in vitro [71].

In addition to its pivotal role in meiosis, MPF serves as a paradigm for cell cycle regulation more broadly, sharing regulatory logic with mitotic control in somatic cells [77,85]. *Xenopus* oocytes thus serve as a model system not only for reproductive biology but also for understanding universal principles of cell cycle transitions.

MPF is activated through a positive feedback loop involving Cdc25-mediated dephosphorylation of Cdk1 and suppression of its inhibitory kinase Wee1. The Mos–MAPK pathway further stabilizes active MPF, ensuring irreversible commitment to GVBD. The activation of MPF not only drives meiotic resumption but also establishes cytoplasmic conditions that are essential for developmental competence [76,80,83,86] (Figure 1B).

### 3.3. GVBD and Ovulation-Inducing Hormones

One of the most distinctive features of vertebrate oocyte maturation is the hormonally induced transition from the prophase I-arrested germinal vesicle (GV) stage to metaphase II. This transition, known as GVBD, marks a pivotal re-entry into meiosis and is tightly regulated by both extracellular signals and intracellular signaling pathways [14,39,40].

In amphibians such as *Xenopus laevis*, progesterone plays a central role in triggering oocyte maturation [15,87]. Produced by surrounding follicle cells under gonadotropin stimulation, progesterone binds to membrane-associated receptors on the oocyte surface [88,89]. This interaction leads to the downregulation of adenylate cyclase activity, reducing cAMP levels and disrupting the PKA-dependent maintenance of meiotic arrest [90].

The reduction in cAMP leads to a cascade of dephosphorylation events and the eventual activation of the Mos-MAPK signaling pathway [41,91], which in turn stabilizes the synthesis and activation of Cdc25, a phosphatase that activates Cdk1. Active MPF levels rise sharply, leading to GVBD [71]. GVBD is characterized cytologically by the dissolution of the nuclear envelope, allowing for the mixing of nuclear and cytoplasmic components and chromatin condensation. This event is readily observable in *Xenopus* oocytes as the central white spot in the animal hemisphere fades [50] (Figure 1B).

GVBD provides an ideal morphological marker to assess hormonal responsiveness and MPF activation in experimental setups [14]. It is often used in in vitro maturation assays to examine the roles of specific hormones, inhibitors, or gene products. Furthermore, *Xenopus* oocytes can be experimentally manipulated to mimic or block GVBD by microinjection of dominant-negative proteins, antisense oligonucleotides, or pharmacological agents [41], making them a highly tractable system for dissecting hormonal signaling cascades.

Understanding GVBD is essential for elucidating the hormonal control of meiotic progression, with implications in reproductive biology, fertility treatments, and endocrine disruption research [92]. The hormonal sensitivity and large cytoplasmic volume of *Xenopus* oocytes make them a powerful system to study steroid signaling and post-transcriptional regulation in vertebrate cells.

Through progesterone-induced MPF activation, the oocyte acquires developmental competence, defined as the ability of the mature egg to undergo fertilization and support normal embryogenesis [25,93,94,95,96] (Figure 1B).

### 3.4. Experimental Detection of MPF Activity In Vitro

The biochemical analysis of MPF activity has played a pivotal role in advancing our understanding of cell cycle regulation, particularly in the context of oocyte maturation. In *Xenopus laevis*, the accessibility and large size of oocytes allow for convenient extraction of cytoplasmic content and precise biochemical manipulation, providing a uniquely powerful model for in vitro assays of kinase activity [78,79].

One of the most widely utilized assays for MPF activity is the histone H1 kinase assay [71,97]. This assay measures the phosphorylation of exogenously added histone H1, a known substrate of the Cyclin B/Cdk1 complex, using radioactive ATP. Extracts from oocytes at various maturation stages (e.g., GV stage, post-progesterone treatment, post-GVBD) are incubated with histone H1 in the presence of [γ-32P]ATP, and the level of H1 phosphorylation serves as a proxy for MPF activity. The assay provides both temporal resolution and semi-quantitative insight into the activation kinetics of Cdk1.

Beyond the histone H1 kinase assay, in vitro maturation systems allow the tracking of oocyte progression through hormonal stimulation, enabling the collection of samples at defined time points [85]. The use of non-radioactive alternatives such as phospho-specific antibodies for Western blotting or in-gel kinase assays has expanded accessibility and safety [80]. More recently, mass spectrometry-based phosphoproteomics has enabled the broader profiling of downstream substrates and pathway interactions during MPF activation [98].

Importantly, these in vitro techniques also allow perturbation experiments such as mRNA microinjection of dominant-negative or constitutively active Cdk1 mutants, pharmacological inhibitors, or antisense oligonucleotides, offering mechanistic insights into the regulation of oocyte maturation [11,20]. The dynamic nature of MPF activation, often exhibiting switch-like behavior, can also be mathematically modeled to explore the bistable characteristics of cell cycle transitions [99,100].

### 3.5. Summary

The oocyte development process in *Xenopus laevis* is marked by a tightly regulated sequence of stages, culminating in the arrest of meiosis at the prophase of the first meiotic division—commonly referred to as the germinal vesicle (GV) stage. This chapter explores the molecular and cellular dynamics underlying this arrest and the critical signaling mechanisms that enable the oocyte to maintain this dormant yet highly poised state.

A central theme is the function of MPF, a heterodimeric complex of Cdk1 and Cyclin B, which governs the resumption of meiosis upon hormonal stimulation, most notably by progesterone. The chapter dissects how MPF activity is stringently suppressed during the GV stage through inhibitory phosphorylation and the sequestration of regulatory proteins such as 14-3-3, Wee1/Myt1, and Cdc25. It further explains how signal transduction cascades—triggered by progesterone binding—ultimately lead to GVBD, marking the reentry into the meiotic cycle.

Practical methodologies for investigating MPF activity, such as in vitro kinase assays using histone H1 as a substrate, are introduced to highlight experimental strategies available for probing meiotic progression. The chapter emphasizes how the *Xenopus* system has contributed significantly to our understanding of cell cycle checkpoints, oocyte maturation timing, and hormonal control.

By combining physiological insights with robust experimental frameworks, this chapter presents the *Xenopus* oocyte as an unparalleled model for dissecting the orchestration of meiosis, hormonal signaling, and oocyte competence.

## 4. Oocyte Maturation and Intracellular Signaling Networks

### 4.1. The Mos-MEK-MAPK Pathway and Activation of MPF

The transition of oocytes from meiotic arrest to maturation is orchestrated by a highly regulated intracellular signaling cascade. Central to this regulation is the Mos-MEK-MAPK pathway, a key player in the activation of MPF, the Cdk1/Cyclin B complex [11,25].

Mos is a serine/threonine kinase whose synthesis is tightly regulated and begins after hormonal stimulation, typically progesterone in *Xenopus* [25]. Upon its translation, Mos initiates a kinase cascade by activating MEK (MAPK/ERK kinase), which in turn phosphorylates and activates MAPK (ERK1/2) [95]. Active MAPK contributes to the stabilization and nuclear localization of Mos itself, thereby forming a positive feedback loop that ensures signal robustness [44,101].

The Mos-MEK-MAPK pathway plays an indispensable role in the full activation of MPF. This is achieved partly through the inhibition of Myt1 and Wee1 kinases, which negatively regulate Cdk1 by phosphorylation, and the activation of Cdc25 phosphatases, which remove inhibitory phosphates from Cdk1 [12]. The resulting dephosphorylated, active MPF drives VBD and entry into metaphase of meiosis I [71,102].

The importance of the Mos-MEK-MAPK pathway is underscored by experiments using antisense oligonucleotides or morpholinos targeting *Mos* mRNA, which prevent MAPK activation and block meiotic progression [25,95]. Conversely, ectopic expression of Mos can induce maturation in immature oocytes even in the absence of hormonal stimuli [44].

Thus, the Mos-MEK-MAPK cascade acts as a molecular bridge between external hormonal cues and the internal cell cycle machinery, ensuring the irreversible and timely activation of meiotic processes in *Xenopus* oocytes [83,102].

### 4.2. Regulation by 14-3-3 Proteins, Wee1/Myt1, and Cdc25

The maturation of *Xenopus laevis* oocytes is orchestrated by a finely tuned regulatory system centered on the activation of MPF, a complex composed of Cdk1 and Cyclin B [78,79]. Crucial to the temporal and spatial control of MPF activation are the regulatory molecules 14-3-3 proteins, Wee1/Myt1 kinases, and Cdc25 phosphatases, which act together in a highly dynamic feedback loop to manage the phosphorylation status and activation state of Cdk1 [83,103].

14-3-3 proteins are highly conserved phosphoserine/threonine-binding proteins that regulate numerous cellular processes by modulating the localization and activity of their binding partners [104]. In immature oocytes, 14-3-3 proteins sequester Cdc25C and other key signaling proteins in the cytoplasm through phosphorylation-dependent interactions [105,106]. This sequestration prevents the premature activation of MPF, thus maintaining the oocyte in prophase I arrest. Upon stimulation by progesterone, key phosphorylation sites on Cdc25 are dephosphorylated, weakening the interaction with 14-3-3 proteins and permitting Cdc25’s translocation into the nucleus [107]. This release from cytoplasmic retention marks a pivotal switch in the signaling cascade leading to MPF activation.

Wee1 and Myt1 are dual-specificity kinases that inhibit Cdk1 activity by phosphorylating it on Tyr15 and Thr14, respectively [108]. In the context of oocyte maturation, these inhibitory phosphorylations are essential for maintaining meiotic arrest. Myt1 is especially important in oocytes, as it is localized to membranes and actively maintains Cdk1 in its inactive state [109]. Progesterone stimulation ultimately leads to the inhibition of Wee1/Myt1 activity, facilitated in part by feedback loops initiated by the small amount of MPF that escapes inhibition [102]. The downregulation of these kinases is essential to permit the accumulation of active MPF, setting the stage for VBD.

Cdc25 phosphatases are responsible for removing the inhibitory phosphates from Cdk1, thereby activating the MPF complex. In *Xenopus* oocytes, Cdc25C is the major isoform involved in meiosis and is tightly regulated by both phosphorylation and localization [110,111]. Once freed from 14-3-3-mediated inhibition, Cdc25C itself becomes a target for phosphorylation by active MPF and the Mos-MEK-MAPK cascade, establishing a robust positive feedback loop [102]. This loop ensures the rapid and irreversible activation of MPF, facilitating the synchronized progression of meiosis.

The interplay between 14-3-3 proteins, Wee1/Myt1 kinases, and Cdc25 phosphatases forms a bistable switch that governs the transition from meiotic arrest to oocyte maturation [83]. This switch-like behavior allows oocytes to remain stably arrested until an appropriate hormonal signal is received, after which the system rapidly transitions to the mature state. Such tight regulation underscores the biological importance of preventing premature oocyte maturation and ensures developmental competency [103].

This regulatory network exemplifies the complexity of signaling coordination during oocyte maturation and highlights potential points of intervention in reproductive biology and developmental research.

### 4.3. Transport Pathways, Localization Control, and Translational Regulation (CPEB and Poly(A) Tail)

In addition to the intricate signaling networks governing meiotic progression, the spatial and temporal regulation of mRNA localization and translation plays a fundamental role in *Xenopus* oocyte maturation [9,23]. These post-transcriptional mechanisms ensure that proteins necessary for maturation, fertilization, and early development are synthesized at the correct time and location within the cell.

*Xenopus* oocytes exhibit highly polarized intracellular organization, where maternal mRNA transcripts are distributed asymmetrically to define future embryonic axes and cell fates [9,13]. Key mRNAs such as Vg1, VegT, and Xwnt11 are transported along the cytoskeleton and localized to the vegetal pole [13,26,43,112,113].

The transport of these transcripts is mediated by RNA-binding proteins and motor proteins, including kinesin and dynein, which direct RNP granules along microtubules and actin filaments [114,115]. Localization elements in the 3′ untranslated region (3′UTR) are critical for proper targeting [116]. These mechanisms not only spatially restrict mRNA function but also regulate translational timing by maintaining mRNAs in a dormant state until needed [13].

One of the hallmark mechanisms controlling translation in *Xenopus* oocytes is cytoplasmic polyadenylation [23,117,118]. Dormant mRNAs possess short poly(A) tails and are translationally repressed in immature oocytes. Upon hormonal stimulation, selected transcripts are activated through poly(A) tail elongation, enabling recruitment of ribosomes.

This regulation is mediated by the cytoplasmic polyadenylation element (CPE) in the 3′UTR, recognized by CPEB (cytoplasmic polyadenylation element-binding protein) [119]. CPEB forms a complex with proteins such as Maskin and PARN, maintaining translational repression [120,121,122].

Progesterone stimulation induces phosphorylation of CPEB, leading to dissociation of Maskin and recruitment of the cytoplasmic poly(A) polymerase GLD-2 [120]. This elongation of the poly(A) tail promotes the binding of poly(A)-binding protein (PABP), resulting in circularization of the mRNA and translational activation [112,118,123].

Cytoplasmic polyadenylation ensures the rapid synthesis of proteins such as Cyclin B1, Mos, and G10, which are critical for MPF activation, spindle assembly, and cortical reorganization [24,124].

This mechanism allows oocytes to respond rapidly to maturation cues without transcription, as the oocyte nucleus is transcriptionally silent at this stage [23]. Thus, translational control through CPEB and poly(A) tail regulation provides spatiotemporal precision, ensuring developmental competence and successful transition to fertilization [125].

### 4.4. Spatiotemporal Dynamics of Oocyte Maturation

Oocyte maturation is not merely a biochemical transition from meiotic arrest to metaphase, but a precisely choreographed cellular process characterized by spatial and temporal regulation of signaling and structural remodeling [11,106]. The concept of spatiotemporal dynamics emphasizes the coordinated interplay between intracellular architecture, signaling cascades, and cytoplasmic reorganization.

Signaling pathways such as Mos-MEK-MAPK and MPF are activated in distinct regions of the oocyte rather than uniformly throughout the cytoplasm [83,126]. For example, MAPK activity has been observed to initiate near the cortex and propagate inward, correlating with cortical rearrangements and cytoskeletal reorganization [127,128].

Similarly, mRNA localization and translational activation are spatially regulated. Dormant transcripts controlled by CPEB-dependent mechanisms are relocalized from the perinuclear region to cortical domains, where translation is initiated [124]. Active MPF, in turn, accumulates in the germinal vesicle prior to GVBD and subsequently spreads throughout the cytoplasm to coordinate spindle assembly [71].

Time-resolved studies have delineated a stereotypical sequence of events following progesterone stimulation [11,129]:

Minutes 0–30: Early translation of *Mos*, activation of MAPK, cytoskeletal priming.

Minutes 30–60: Positive feedback amplifies MPF activity; GVBD occurs.

Minutes 60–90: Chromosome condensation, meiotic spindle formation, spindle migration to the animal pole.

This cascade reflects the bistable and switch-like dynamics of the oocyte cell cycle, where once MPF activation reaches a threshold, meiotic progression becomes irreversible [126].

Maturation is accompanied by dramatic structural changes: GVBD and chromatin condensation [14]. Reorganization of the microtubule network into a meiotic spindle [130]. Actin cytoskeleton remodeling to support spindle migration and cortical granule positioning [131]. Mitochondrial relocalization to provide localized ATP supply during high energy demand [132]. These spatial rearrangements are not passive consequences but also feed back into signaling processes, reinforcing the timing and robustness of maturation.

The spatiotemporal coordination of molecular signaling and structural reorganization transforms a dormant oocyte into a fertilization-competent egg. This systemic view highlights how biochemical switches, cytoskeletal dynamics, and translational regulation converge to enable precise cellular decision-making [129,133].

Future advances—particularly live-cell imaging, biosensor technologies, and computational modeling—are expected to unravel how these dynamics are encoded and conserved across species, broadening insights into universal principles of cell cycle regulation [134].

### 4.5. Summary

The resumption and completion of oocyte maturation in *Xenopus laevis* are orchestrated by a cascade of intracellular signaling events that converge on the activation of MPF. This chapter examines the molecular circuits responsible for initiating and sustaining this maturation process, with a focus on cytoplasmic signaling networks that regulate both temporal and spatial aspects of cellular activation.

At the heart of this process is the Mos-MEK-MAPK pathway, which serves as a critical positive regulator of MPF activation. The synthesis and stabilization of Mos protein—induced by progesterone—triggers a kinase cascade that activates p42MAPK and ultimately contributes to MPF amplification. In parallel, the regulation of MPF-inhibitory pathways involves complex interactions among 14-3-3 proteins, Wee1/Myt1 kinases, and the dual-specificity phosphatase Cdc25, forming a bistable switch that ensures the irreversible transition to meiotic resumption.

Another layer of control is provided by the localization and translation of maternal mRNAs, especially those regulated by cytoplasmic polyadenylation. Proteins such as CPEB and the poly(A) polymerase machinery modulate the timing of mRNA translation, coordinating protein synthesis with meiotic progression. Additionally, the dynamic redistribution of signaling molecules and organelles across the animal-vegetal axis exemplifies the spatial regulation of maturation signals.

This chapter also introduces the concept of “spatiotemporal dynamics” in oocyte maturation, highlighting how coordinated signaling events across time and intracellular space ensure precise control over meiotic progression. By integrating biochemical, molecular, and imaging-based insights, the *Xenopus* model offers a powerful platform for elucidating the universal logic of cellular decision-making during maturation.

## 5. The Moment of Fertilization—Calcium Waves, Blocks, and Activation

### 5.1. Sperm Guidance and Adhesion: Mechanisms and Species Specificity

Fertilization in externally fertilizing species such as *Xenopus laevis* relies on a sequence of highly regulated events, beginning with the guidance and adhesion of spermatozoa to the oocyte. These steps are essential not only for successful fertilization but also for ensuring species specificity and preventing polyspermy [135,136].

Following ovulation, the oocyte is surrounded by the vitelline envelope and a layer of follicle cells that secrete chemoattractants. These signaling molecules create a gradient that guides spermatozoa toward the egg [137,138]. In *Xenopus*, the oocyte jelly coat contains glycoproteins and peptides, such as oligopeptides derived from the glycoprotein gp69/70, that act as chemotactic cues [138,139].

Motile sperm exhibit chemotaxis, adjusting their flagellar beat patterns to navigate up the chemoattractant gradient. This process involves calcium influx through CatSper channels in the sperm tail, leading to modulation of swimming behavior [140,141,142,143]. Though more prominently characterized in mammals and sea urchins, such mechanisms are conserved across many species [144].

Once sperm reach the egg, the next critical step is recognition and adhesion to the egg envelope. Species specificity is largely mediated by interactions between sperm surface proteins and components of the vitelline envelope [145]. In *Xenopus laevis*, the glycoprotein gp41 has been implicated in mediating species-specific sperm binding [89].

Lectin-like domains and glycan moieties on the envelope are thought to form complementary binding sites for sperm surface molecules [146]. Cross-species fertilization experiments have demonstrated that the vitelline envelope serves as a barrier to heterologous sperm, supporting the idea of molecular signatures that ensure species fidelity [147].

Upon binding, sperm undergo an acrosome reaction-like process, enabling the release of enzymes and exposure of fusogenic domains. Although *Xenopus* sperm lack a true acrosome, they exhibit similar behavior through exocytosis of membrane-bound vesicles and cytoskeletal rearrangements [148,149].

Egg membranes also become primed for fusion via reorganization of microvilli and clustering of egg membrane proteins such as integrins and tetraspanins [150,151,152,153]. The localized calcium-rich microenvironment near the site of sperm contact is thought to promote membrane fusion competence [17].

Understanding the molecular details of sperm guidance and binding in *Xenopus* provides insight into the conserved and divergent strategies of fertilization across species [21,154]. It also sheds light on mechanisms ensuring reproductive isolation and fidelity, offering translational relevance to reproductive biotechnology and contraception [155].

The process of sperm-egg interaction in *Xenopus* thus reflects a delicately orchestrated dialog of molecular recognition, cellular navigation, and membrane dynamics that sets the stage for the cascade of fertilization-induced activation events.

### 5.2. Sperm-Derived PLCζ and the IP3-Mediated Fertilization Calcium Wave

The entry of a sperm into the oocyte is the trigger for a series of highly orchestrated intracellular events, most notably the initiation of a transient yet robust increase in intracellular calcium concentration. This calcium signal, manifesting as a wave that traverses the egg cytoplasm, is essential for egg activation and the initiation of embryonic development [156,157] (Figure 1C).

Upon fusion of the sperm and egg membranes, a sperm-derived soluble factor, phospholipase C zeta (PLCζ), is released into the oocyte cytoplasm [158,159]. PLCζ is a sperm-specific isoform of phospholipase C that is both necessary and sufficient to trigger the calcium release in eggs across multiple vertebrate species [160,161], but enigmatically, not in *Xenopus laevis*. The absence of a PLCZ1 ortholog in amphibians indicates that *Xenopus* relies on distinct mechanisms for triggering egg activation [162]. Instead of PLCζ-dependent signaling, *Xenopus* eggs appear to utilize protease-mediated pathways in which a sperm-derived trypsin-like protease cleaves the egg-membrane receptor uroplakin III (UPIII) to initiate Src-family kinase activation, with MMP-2 further contributing to localized signal propagation [163,164].

PLCζ functions by hydrolyzing phosphatidylinositol 4,5-bisphosphate (PIP2), a membrane phospholipid, into two second messengers: inositol 1,4,5-trisphosphate (IP3) and diacylglycerol (DAG). Among these, IP3 plays the pivotal role in initiating calcium release [165].

IP3 binds to its receptor (IP3R), a ligand-gated calcium channel localized on the endoplasmic reticulum (ER) membrane [166,167]. Binding of IP3 induces the opening of these channels, allowing the release of calcium ions from ER stores into the cytoplasm.

The initial calcium increase further activates adjacent IP3Rs in a regenerative process known as calcium-induced calcium release (CICR), thereby generating a wave that propagates from the sperm entry point across the egg [168,169].

In *Xenopus* eggs, the ER is organized into a network of dense tubules and vesicles that are particularly abundant in the cortex, facilitating rapid and widespread calcium mobilization [170]. The wave propagates at a speed of approximately 5–10 µm/sec and lasts for several minutes before returning to basal levels [171].

The fertilization-induced calcium transient has multiple downstream effects, including the resumption of the cell cycle, cortical granule exocytosis to block polyspermy, and activation of developmental gene expression [156,157] (Figure 1C).

Experimental microinjection of recombinant PLCζ or IP3 into unfertilized *Xenopus* eggs mimics natural fertilization-induced calcium signals, confirming the sufficiency of this pathway [159,160]. Moreover, eggs lacking IP3R function fail to initiate activation even after sperm entry, underscoring the necessity of this signaling cascade [167].

The conservation of PLCζ-triggered calcium signaling across species highlights a deeply rooted evolutionary mechanism for egg activation [159,172,173]. However, recent work in teleost fish demonstrates that another non-mammalian vertebrate species can also employ protease-activated receptor signaling in place of PLCζ. In zebrafish, Ma and Carney [174] showed that sperm-derived proteases activate the protease-activated receptor Par2 (PAR2) on the egg surface, eliciting a propagating calcium wave that supports early blastomere cleavage. This mechanism closely parallels the protease–receptor pathways observed in *Xenopus* and suggests that proteolytic activation of egg-membrane receptors may represent an evolutionarily widespread strategy among vertebrates lacking PLCζ.

Understanding these processes in *Xenopus* not only elucidates fundamental principles of fertilization but also informs assisted reproductive technologies (ART), particularly in cases of sperm-related infertility [175]. The fertilization calcium wave thus represents a central molecular event, linking the external cue of sperm fusion to the internal reprogramming of the oocyte into a zygote capable of development.

In addition to the well-characterized IP3-dependent fertilization calcium wave, many externally fertilizing species exhibit a rapid, transient “cortical flash,” a brief Ca^2+^ influx that precedes the global wave. This event has been described most extensively in echinoderms, including sea urchins and starfish, where it is considered essential for proper downstream activation events and early embryonic development. While a prominent cortical flash has not been consistently observed in *Xenopus*, placing the frog system in comparative context highlights how fertilization-triggered Ca^2+^ dynamics vary across metazoans and emphasizes the conserved logic of Ca^2+^-driven egg activation [157,176,177,178].

### 5.3. Sperm-Derived Trypsin-like Protease and Egg UPIII–Src Tyrosine Kinase Axis

Beyond the well-characterized calcium wave triggered by sperm-derived PLCζ, recent studies have revealed an additional signaling axis that plays a pivotal role in the initiation of egg activation: the interaction between a sperm surface trypsin-like protease and the egg membrane protein uroplakin 3 (UPK3), leading to the activation of Src family tyrosine kinases [179,180,181,182]. Importantly, because *Xenopus* seems to lack a PLCZ1 ortholog and does not rely on PLCζ for initiating Ca^2+^ signaling, protease-mediated membrane signaling represents a primary rather than auxiliary pathway for egg activation in amphibians.

Spermatozoa carry a repertoire of proteolytic enzymes on their surface, including serine proteases with trypsin-like activity [146,183]. In *Xenopus laevis*, a specific sperm-derived protease has been implicated in cleaving or interacting with egg surface proteins in a highly regulated, species-specific manner.

This proteolytic interaction is proposed to be among the earliest molecular events post-sperm binding, preceding or complementing the activity of PLCζ [181]. Recent findings in teleost fish support the broader significance of protease-driven egg activation among non-mammalian vertebrates. In zebrafish, Ma and Carney [174] demonstrated that sperm-derived proteases cleave and activate the protease-activated receptor Par2 (PAR2), inducing a robust Ca^2+^ wave that supports early blastomere cleavage. This mechanism—protease activation of an egg-membrane receptor—parallels the UPIII–Src axis in *Xenopus* and suggests that proteolytic signaling may represent an evolutionarily conserved strategy in vertebrates lacking PLCζ.

UPIII, originally characterized in the urothelium, is expressed on the *Xenopus* egg plasma membrane and has emerged as a key candidate receptor for sperm-borne signals [179,180,181,184]. UPIII is structurally poised to transmit extracellular cues into intracellular responses. Binding or cleavage by the sperm trypsin-like protease induces a conformational change or release of an inhibitory domain, enabling downstream signaling events [182]. The interaction between sperm protease and UPK3 triggers the activation of Src family kinases (SFKs), particularly c-Src, within minutes after sperm-egg contact [185]. This kinase activity leads to rapid phosphorylation of specific targets in the egg cortex and may synergize with or augment the PLCζ/IP3-induced calcium signaling cascade. The involvement of Src is supported by pharmacological inhibition studies, where SFK inhibitors delay or prevent egg activation [181,186,187,188].

While the PLCζ/IP3 pathway is largely sufficient for initiating calcium release and downstream activation, the UPK3–Src axis likely serves as a parallel or reinforcing pathway [185]. It may contribute to spatial fidelity and robustness of the fertilization response, particularly in the rapid activation of cortical granule exocytosis and cytoskeletal rearrangement [182].

The discovery of protease-mediated egg surface signaling adds a novel layer of complexity to our understanding of fertilization [189]. It also raises the possibility that redundant or synergistic pathways ensure successful activation under diverse physiological conditions. The UPK3–Src axis in *Xenopus* may have analogs in mammalian systems, offering potential translational insights into fertilization failure and therapeutic targets in ART [172,184]. This section underscores the emerging view that fertilization is not governed by a single pathway but by an interwoven network of biochemical signals, of which sperm proteases and egg tyrosine kinases are integral components.

Fertilization in mammals is initiated by highly specific molecular interactions at the gamete plasma membranes. The sperm protein IZUMO1 binds to its oocyte receptor JUNO, mediating species-specific recognition and adhesion [190,191]. Structural studies revealed that JUNO evolved from a folate receptor to function as an adhesion molecule [192,193], while IZUMO1 forms large multiprotein complexes and redistributes during the acrosome reaction [191,194,195]. On the oocyte side, the tetraspanin CD9 is essential for maintaining microvillar architecture and promoting membrane fusion [152,196,197,198]. Reviews highlight how these proteins collectively orchestrate sperm–egg binding and fusion [199,200].

More recently, AI-based structural prediction and molecular simulations have accelerated the discovery of novel sperm factors. AlphaFold-Multimer predicted an IZUMO1–SPACA6–TMEM81 complex, and genetic knockout confirmed TMEM81 as indispensable for fertilization in mice. This was reported as a conserved fertilization complex bridging sperm and egg across vertebrates [201] and discussed in news coverage [202]. Furthermore, supercomputer-based simulations successfully modeled the dynamic JUNO–IZUMO1 complex, providing detailed insights into pre-fusion protein dynamics [203]. Together, these studies reveal how classical molecular genetics and modern AI-driven predictions converge to elucidate the mechanisms of gamete recognition and fusion.

In mammals such as the mouse, the prevailing model of oocyte activation centers on the sperm-specific enzyme phospholipase C zeta (PLCζ), which triggers intracellular Ca^2+^ oscillations after gamete fusion. This mechanism is supported by genetic, biochemical, and imaging studies and is considered essential for initiating embryo development.

By contrast, in *Xenopus laevis*, genomic analyses indicate the absence of PLCZ1 orthologs, suggesting that amphibians employ distinct strategies for fertilization-triggered egg activation [162]. Experimental studies have demonstrated that sperm proteases are crucial: early work showed that *Xenopus* egg activation can be induced by sperm-derived proteolytic activity [183]. Trypsin-like proteases partially cleave the egg membrane protein UPIII, a novel Src substrate in oocyte membrane microdomains, thereby activating Src tyrosine kinase and downstream Ca^2+^ signaling [179,181,182]. This UPIII–Src signaling module becomes functional during oocyte maturation and mediates bidirectional gamete signaling [181].

Another line of evidence points to the role of sperm MMP-2. Studies in *Xenopus tropicalis* revealed that sperm-surface MMP-2 is indispensable for the fast electrical block to polyspermy, functioning immediately after gamete contact [163,164]. Thus, in amphibians, protease-mediated remodeling of the egg membrane and activation of egg tyrosine kinase pathways substitute for PLCζ-mediated Ca^2+^ oscillations observed in mammals.

Collectively, these findings highlight that while mammals rely on PLCζ as the physiological sperm factor, amphibians such as *Xenopus* utilize protease-dependent signaling mechanisms (MMP-2, trypsin-like proteases, UPIII–Src) for oocyte activation. The absence of PLCζ in amphibian genomes underscores the evolutionary divergence of fertilization strategies. The precise molecular intermediates and the extent to which these pathways operate independently or synergistically remain under active investigation. Further comparative analyses across vertebrates indicate that protease-mediated activation may take diverse molecular forms while fulfilling a conserved functional role in initiating egg activation.

### 5.4. Formation of the Fertilization Envelope, Vitelline Layer Modification, and Polyspermy Block Mechanisms

Following the calcium wave triggered at fertilization, the egg undergoes a suite of structural and biochemical changes to prevent polyspermy and ensure successful development. In *Xenopus laevis*, these changes involve rapid modifications to the vitelline envelope, the formation of a fertilization envelope, and the hardening of the egg surface [147,204].

#### 5.4.1. Cortical Granule Exocytosis and Fertilization Envelope Formation

The intracellular calcium surge induces exocytosis of cortical granules located just beneath the egg plasma membrane [205,206]. These granules release enzymes and structural proteins into the perivitelline space. Proteases cleave proteins that tether the vitelline envelope to the egg membrane, facilitating its elevation and transformation into the fertilization envelope [145]. This envelope acts as a physical barrier to further sperm entry [207].

#### 5.4.2. Vitelline Layer Hardening and Egg Yolk Plug Consolidation

Among the released enzymes are peroxidases and transglutaminases that crosslink proteins in the vitelline layer, resulting in biochemical hardening [208]. Additionally, changes in egg surface glycoproteins and ion fluxes contribute to this stiffening [21,147,154,209,210]. The yolk plug, a region of concentrated yolk material, undergoes structural rearrangement and compaction, which further reinforces the egg’s integrity [205].

*Xenopus* eggs utilize both fast and slow blocks to polyspermy. The fast block involves a transient depolarization of the egg membrane upon sperm-egg fusion, preventing additional sperm binding [211]. This is followed by the slower, irreversible modification of the vitelline envelope described above [145]. These layers of protection are essential, as polyspermy leads to catastrophic mitotic errors and embryonic lethality [207].

The remodeling of the egg’s extracellular matrix not only blocks additional sperm but also contributes to species specificity. Proteolytic modification alters sperm-binding sites, ensuring that only conspecific sperm can initially interact, and that no further sperm entry occurs once fertilization is achieved [21,147,207,208,209,210,212].

#### 5.4.3. Functional Significance and Research Applications

The study of fertilization envelope formation and polyspermy blocks has shed light on membrane fusion, extracellular matrix remodeling, and calcium-regulated secretion [147]. It also has practical implications in assisted reproductive technologies, where controlling sperm number and timing is crucial [172]. Together, these mechanisms illustrate how the fertilized egg rapidly transforms itself into a protected, developmentally competent entity, highlighting the sophistication of early developmental regulation in vertebrates.

Although the elevation of the fertilization envelope and the accompanying mechanical block are traditionally presented as robust and highly effective, recent studies in echinoderms have demonstrated that neither the mechanical nor the electrical block is absolutely fail-safe. In starfish, for example, supernumerary sperm can enter the egg even after full elevation of the fertilization envelope. Similarly, in sea urchin eggs, the electrical block to polyspermy exhibits species-specific variability and, under certain experimental conditions, can be bypassed. These findings underscore that polyspermy-block mechanisms, while generally reliable, operate within a biological continuum rather than as strictly binary systems. Integrating these insights provides a more nuanced understanding of how *Xenopus* and other metazoan embryos mitigate the risk of polyspermy [21,213].

### 5.5. Resumption of the Cell Cycle After Activation—Inactivation of CSF

In *Xenopus laevis*, mature eggs are arrested at metaphase of meiosis II due to the action of CSF, a molecular complex that maintains the cell in a suspended state of division readiness [25,79]. Fertilization provides the cue to release this block and trigger cell cycle resumption, culminating in the onset of embryonic development.

The maintenance of meiotic metaphase arrest is controlled by high activity of MPF, composed of Cdk1 and Cyclin B [77]. CSF enforces this arrest by preventing Cyclin B degradation, thus sustaining MPF activity. The primary molecular constituent of CSF is the Mos-MAPK-p90Rsk pathway, which stabilizes Cyclin B levels and inhibits the anaphase-promoting complex/cyclosome (APC/C) [25,214].

Fertilization-induced calcium elevation plays a pivotal role in initiating cell cycle progression [157]. The surge activates calmodulin-dependent protein kinase II (CaMKII), which in turn triggers degradation of Emi2 (early mitotic inhibitor 2), a potent APC/C suppressor [215]. This allows APC/C to polyubiquitinate Cyclin B, marking it for proteasomal degradation and consequently inactivating MPF [216].

As MPF activity declines, the egg completes meiosis II, extrudes the second polar body, and forms a haploid female pronucleus [39,40]. Concurrently, the sperm contributes its nucleus and centriole, forming the zygote. The zygotic nucleus then enters the first embryonic mitosis, guided by centrosomal microtubule organization derived from the sperm [217].

Understanding CSF inactivation mechanisms illuminates general principles of cell cycle regulation and fertilization control [92]. Clinically, manipulating this arrest-release balance offers strategies in assisted reproduction, oocyte activation protocols, and somatic cell nuclear transfer [22]. The transition from a quiescent, arrested egg to an actively dividing zygote is one of the most rapid and tightly regulated switches in developmental biology, and *Xenopus* continues to serve as an indispensable model in decoding its molecular choreography [157].

### 5.6. Summary

Fertilization marks a pivotal turning point in the developmental timeline of *Xenopus laevis*, transforming a mature but arrested oocyte into a developmentally competent zygote. This chapter explores the molecular mechanisms by which sperm entry triggers a cascade of signaling events, leading to egg activation and the re-entry into the cell cycle.

Species-specific recognition at the egg surface initiates the fertilization process, involving coordinated interactions between sperm-derived proteases and egg membrane proteins such as UPIII, which subsequently activate the Src family tyrosine kinases. One of the earliest and most critical outcomes of fertilization is the generation of a fertilization-induced calcium wave, a cytoplasmic surge initiated by sperm-delivered PLCζ that produces IP3 and triggers intracellular Ca^2+^ release through IP3 receptors. This calcium transient not only mediates egg activation but also coordinates multiple downstream events required for embryogenesis.

Among these downstream events is the cortical reaction, which establishes a fertilization envelope and hardens the vitelline layer—mechanisms crucial for preventing polyspermy. The calcium wave also triggers the inactivation of CSF, allowing the oocyte to exit metaphase II arrest and resume the cell cycle through MPF inactivation.

## 6. Fertilization as a Trigger for Early Development and the Regulation of Activation Factors

### 6.1. Xenopus in the Context of Mosaic Versus Regulative Development

One of the most compelling aspects of developmental biology is how multicellular organisms orchestrate the formation of complex body plans from a single-celled zygote. Two fundamental paradigms—mosaic and regulative development—define the spectrum of developmental strategies. Mosaic development implies that each cell inherits a distinct set of determinants that irrevocably commits it to a specific fate [218,219]. In contrast, regulative development allows for flexibility in cell fate through intercellular signaling and positional information [220,221].

*Xenopus* laevis occupies a central position in the study of regulative development. Despite exhibiting elements of pre-patterned asymmetry, such as animal-vegetal polarity and the localization of maternal mRNAs (e.g., VegT, Vg1) [13,27,43,222,223], *Xenopus* embryos demonstrate significant regulative potential. For instance, if blastomeres are experimentally separated at the 2-cell stage, they can independently generate complete embryos, reflecting an ability to re-establish body axes and tissue proportions [224].

Classic transplantation experiments—such as those conducted by Spemann and Mangold—demonstrated the existence of an organizer region in *Xenopus* that governs the patterning of the embryo through inductive interactions [225]. This finding was pivotal in establishing regulative development as a dominant theme in vertebrates.

Moreover, studies involving the redistribution of cytoplasmic determinants via UV irradiation or centrifugation illustrate the embryo’s plasticity and ability to compensate for disrupted cues, further emphasizing its regulative character [226,227,228,229].

#### A Contextual View

*Xenopus* therefore serves as a versatile system that blends elements of both developmental paradigms [221,230]. While certain aspects are reminiscent of mosaic development due to maternal determinants, its capacity for axis re-specification, inductive signaling, and cell–cell communication aligns it more closely with regulative models, particularly during the cleavage and blastula stages.

This duality renders *Xenopus* an invaluable organism for dissecting the molecular and cellular basis of developmental plasticity and robustness [221,230].

### 6.2. Cytoplasmic Determinants and the Initiation of Mesoderm Induction

Early *Xenopus* embryogenesis is profoundly influenced by localized maternal cytoplasmic factors that establish the initial conditions for germ layer formation. Among these, VegT, Vg1, and β-catenin play central roles in initiating the signaling cascades that drive mesoderm induction [4,231,232,233].

VegT is a T-box transcription factor localized to the vegetal hemisphere of the oocyte [223]. Following fertilization, VegT mRNA is translated and induces the expression of nodal-related genes, which are crucial for mesoderm and endoderm formation [234,235,236]. Depletion of VegT results in the loss of both germ layers, underscoring its essential role in early patterning [237].

Vg1, a member of the TGF-β superfamily, is similarly localized to the vegetal cortex [26]. Although maternally deposited as an mRNA, Vg1 becomes translationally active during early development and synergizes with VegT-induced signals to enhance nodal expression and downstream mesodermal gene activation [238,239].

β-catenin plays a distinct but complementary role. Following cortical rotation triggered by fertilization, β-catenin becomes stabilized in the dorsal region of the embryo [240,241]. This accumulation specifies the Nieuwkoop center—a dorsal vegetal signaling hub that promotes organizer formation and dorsal mesoderm specification [28,242]. β-catenin functions via the Wnt pathway, activating genes such as *siamois* and *twin* that define the dorsal axis [243,244].

Together, these cytoplasmic factors orchestrate a highly coordinated inductive network that sets the stage for germ layer segregation. The interplay between vegetal signals (VegT, Vg1) and dorsal determinants (β-catenin) exemplifies the integration of spatial and temporal cues in embryonic pattern formation [245,246].

These insights provide a mechanistic foundation for understanding how a seemingly homogeneous cytoplasm is transformed into a spatially complex signaling environment capable of initiating vertebrate body plan development.

### 6.3. Germ Layer Differentiation Programs and the Onset of Translation

The process of germ layer differentiation in *Xenopus* embryos—namely, the formation of ectoderm, mesoderm, and endoderm—relies on tightly controlled temporal and spatial regulation of mRNA translation. Although maternal mRNAs are stockpiled in the oocyte, their translation is temporally repressed until appropriate cues are received during fertilization and early cleavage stages [23,118].

One of the key regulators of translational initiation is CPEB, which mediates the polyadenylation and activation of specific mRNAs, such as those encoding VegT and Vg1 [122,247]. This activation ensures that essential transcription factors and signaling molecules are synthesized at the correct developmental window.

CPEB-regulated transcripts containing cytoplasmic polyadenylation elements (CPEs) in their 3′ untranslated regions are subject to elongation of their poly(A) tails upon fertilization or during specific developmental timepoints, allowing for translation initiation [117,119]. Pumilio and Maskin are additional proteins that contribute to the temporal repression of these transcripts before polyadenylation [248,249].

Each germ layer lineage is characterized by a unique set of maternal and zygotic mRNAs whose translation is modulated in a stage-specific manner. For example: Endodermal specification is reinforced by the translation of *VegT* and *Sox17* [237,250]. Mesodermal induction is driven by the combined activity of *Vg1*, *Nodal*, and *Mix1* family members [238,246]. Ectodermal maintenance involves inhibition of mesoderm-inducing signals via factors such as *Coco* and *Dand5* [251,252]. These differentiation programs are also influenced by intracellular localization of determinants, nuclear import of transcription factors, and signal transduction cascades initiated by cell–cell interactions [4].

A major developmental milestone known as the mid-blastula transition (MBT) marks the handover from maternal to zygotic control [73,74]. During this period, zygotic transcription becomes robust, and translational repression of maternal transcripts is gradually relieved or actively degraded [253].

Understanding the intricacies of translational regulation during early *Xenopus* development offers a powerful lens into how developmental timing, spatial organization, and cell fate commitment are coordinated to establish body plan architecture [245,254].

### 6.4. Sperm-Derived Centrosome and the Establishment of the First Cleavage Axis

In *Xenopus laevis*, the sperm not only delivers paternal DNA during fertilization but also contributes a critical organelle: the centrosome. Unlike many other species, *Xenopus* oocytes do not contain functional centrioles [217,255]. Therefore, the sperm-derived centriole becomes the foundation for the zygotic centrosome, which orchestrates the microtubule organizing center (MTOC) necessary for proper mitotic spindle formation and cleavage divisions [256].

Following fertilization, the sperm centrosome rapidly nucleates microtubules, establishing a radial aster that guides the sperm nucleus and associated centriole toward the egg center [54,255]. This movement is essential not only for pronuclear apposition but also for the determination of the first cleavage plane. The site where the sperm enters the egg predicts the dorsal-ventral axis due to subsequent cortical rotation [228].

The positioning of the centrosome plays a pivotal role in the alignment of the mitotic spindle during the first division [257]. As the spindle forms, the cleavage furrow bisects the egg through the animal-vegetal axis, typically orthogonal to the sperm entry point.

A hallmark event in *Xenopus* development is the ~30-degree cortical rotation that occurs post-fertilization [258]. This microtubule-dependent movement displaces vegetal cortex components, resulting in the asymmetrical distribution of cytoplasmic determinants, including Wnt signaling activators such as Dishevelled and GBP (GSK3-binding protein) [259,260].

These relocated molecules accumulate in the prospective dorsal side and initiate the Wnt/β-catenin signaling cascade, which is crucial for dorsal axis specification [240,241]. The coordinated interplay between sperm centrosome positioning and cortical rotation effectively links fertilization with the establishment of embryonic polarity [261].

Microsurgical experiments that manipulate the sperm entry point or block cortical rotation confirm the importance of these events in axis formation [228,258]. Disruption of sperm centrosome function or rotation impairs dorsal structure development, resulting in ventralized or radialized embryos [255,262].

The contribution of the sperm centrosome to embryogenesis underscores a broader concept in developmental biology: fertilization is not merely a genetic fusion event but a spatially orchestrated biological trigger that initializes the complex architecture of the embryo.

### 6.5. Summary

Following fertilization and egg activation, *Xenopus laevis* embryos rapidly initiate pattern formation through asymmetric distribution of maternal determinants and cytoplasmic rearrangements. This chapter delves into the earliest stages of embryonic organization, focusing on how spatial cues derived from the egg’s cytoplasm establish the primary body axes and initiate germ layer differentiation.

One of the defining features of *Xenopus* embryogenesis is its regulative nature—contrasting with the mosaic development observed in other organisms such as *C. elegans*. This capacity for regulative development is enabled by localized cytoplasmic determinants that are redistributed following sperm entry. Rotation of the cortical cytoplasm relative to the inner cytoplasm leads to symmetry breaking, with the sperm-derived centrosome playing a central role in orienting the first cleavage plane.

Key maternal factors—including β-catenin, Vg1, and VegT—are differentially localized within the vegetal and dorsal cytoplasm. These molecules act in concert to initiate mesoderm induction, axis specification, and germ layer segregation. Their translation is temporally regulated and spatially restricted through mechanisms such as cytoplasmic polyadenylation and selective mRNA stabilization, ensuring that signaling cascades such as Wnt and TGF-β pathways are activated at the appropriate developmental windows.

The chapter emphasizes how these molecular gradients and regionalized signaling events lay the groundwork for gastrulation and subsequent organogenesis. By dissecting these early patterning mechanisms, researchers gain insight not only into vertebrate embryology but also into broader principles of cell fate determination and tissue organization.

To reflect current knowledge in fertilization biology, recent updates from comparative embryology have been incorporated, including insights summarized in Barresi & Gilbert [176]. These contemporary perspectives revise several long-standing assumptions about Ca^2+^ dynamics and polyspermy block mechanisms, highlighting cross-species diversity and emphasizing that multiple regulatory layers act together to secure successful monospermic fertilization.

By integrating updated models of Ca^2+^ signaling, revised views on polyspermy block mechanisms, and comparative insights from echinoderm and amphibian systems, this review provides a broader and more accurate framework for understanding oocyte activation across metazoans. The *Xenopus* model, enriched by the addition of new schematics and comparative tables, continues to serve as a powerful platform for dissecting the conserved principles governing oocyte maturation, fertilization, and the post-ovulatory fate of unfertilized eggs.

## 7. Post-Ovulatory Fate of Oocytes—The Journey of Unfertilized Eggs

### 7.1. Apoptosis vs. Atresia—Natural Death Pathways of Unfertilized Eggs

Following ovulation, oocytes that fail to undergo fertilization face a programmed fate of cellular demise. Two primary pathways define this process: apoptosis, a genetically encoded cell death mechanism, and atresia, a broader term encompassing degenerative processes that eliminate non-fertilized or developmentally incompetent oocytes.

Apoptosis in *Xenopus* oocytes is characterized by hallmarks such as chromatin condensation, DNA fragmentation, membrane blebbing, and activation of caspase enzymes, particularly caspase-3 [10,263]. Studies have shown that, in the absence of fertilization, matured oocytes eventually activate intrinsic apoptotic pathways. The mitochondrial release of cytochrome c triggers apoptosome formation, leading to caspase activation and cellular dismantling [264]. External stressors, including prolonged in vitro culture or oxidative stress, can accelerate this apoptotic response [265]. Additionally, fertilization is thought to provide survival signals that counteract these death pathways, indicating a fertilization-dependent switch between life and death.

In contrast to apoptosis, atresia refers to the degeneration and resorption of oocytes before or after ovulation through a more heterogeneous set of mechanisms [266,267]. In *Xenopus*, atretic follicles exhibit variable morphological and biochemical changes, often lacking the canonical features of apoptosis. Granulosa cell interactions, hormonal milieu, and follicular environment influence the onset and progression of atresia [268,269]. Atresia may involve autophagic pathways, lysosomal degradation, and non-caspase-dependent cell death mechanisms [270]. This distinction is particularly relevant when examining ovarian tissue rather than isolated ovulated oocytes [269,271,272,273,274,275].

#### Comparative Perspectives and Biological Implications

Understanding the dichotomy between apoptosis and atresia is essential for interpreting the fate of oocytes in both physiological and pathological contexts. In mammals, similar pathways regulate ovarian reserve and fertility [108,266,276], whereas in amphibians like *Xenopus*, the transparent and accessible oocytes allow for detailed mechanistic studies [264,277]. The regulation of oocyte death has implications for reproductive biology, assisted reproductive technologies, and developmental competence assessment. Whether the oocyte dies quietly through apoptosis or is reabsorbed via atresia, its fate reflects a tightly orchestrated decision by the cellular machinery in the absence of sperm entry.

### 7.2. Caspase-Dependent and -Independent Pathways—Mitochondrial Contributions to Oocyte Demise

The molecular mechanisms underlying oocyte death are multifaceted, involving both caspase-dependent and caspase-independent pathways. In *Xenopus* oocytes, caspase-3 plays a pivotal role in executing the apoptotic program, typically activated via the intrinsic mitochondrial pathway [263]. Upon mitochondrial outer membrane permeabilization (MOMP), cytochrome c is released into the cytosol, where it promotes apoptosome assembly with Apaf-1 and initiator caspase-9, ultimately leading to caspase-3 activation [10,264,278,279,280].

However, studies also indicate the presence of caspase-independent mechanisms. These include the release of apoptosis-inducing factor (AIF) and endonuclease G from mitochondria, which translocate to the nucleus and contribute to DNA degradation and chromatin condensation without involving caspase activation [281,282,283]. These alternative routes highlight the versatility of cell death programs and their capacity to ensure oocyte clearance even in the presence of caspase inhibition.

The mitochondrion acts as a central hub in both death pathways, not only as a reservoir of pro-apoptotic factors but also as a metabolic and redox sensor. Changes in mitochondrial membrane potential (Δψm), reactive oxygen species (ROS) production, and ATP depletion are closely associated with oocyte quality and survival [284,285,286,287,288].

Recent work also emphasizes the role of Bcl-2 family proteins in regulating mitochondrial integrity in *Xenopus* oocytes. Pro-apoptotic members such as Bax and Bak promote MOMP, while anti-apoptotic members like Bcl-xL confer resistance to cell death [289,290,291]. The balance between these opposing forces is critical in determining the timing and mode of oocyte demise.

In summary, the mitochondrion not only initiates and executes apoptotic signals but also integrates upstream cues related to oocyte maturation, fertilization status, and environmental conditions. Dissecting the interplay between caspase-dependent and -independent pathways will further elucidate the molecular choreography of oocyte attrition.

### 7.3. Post-Maturation Lifespan Limitation and Molecular Changes in Unfertilized Eggs

In *Xenopus laevis*, matured oocytes that fail to undergo fertilization are subject to intrinsic time limits on their viability, often culminating in programmed cell death [37,253,292,293]. This phenomenon reflects a tightly regulated process involving a cascade of molecular changes that ensure elimination of aged, non-viable gametes.

After maturation, the oocyte is arrested in metaphase II by the activity of the CSF, primarily sustained by the MAPK pathway [20,25]. In the absence of fertilization-induced calcium influx, which triggers the inactivation of CSF and MPF, the oocyte remains in this arrested state. However, prolonged metaphase II arrest leads to the gradual loss of MPF and MAPK activity through protein degradation and reduced kinase stability, undermining the cell’s structural integrity [92].

Concurrently with these changes, oxidative stress accumulates due to elevated levels of reactive oxygen species (ROS), often originating from mitochondria under metabolic strain [37,38,294]. This oxidative stress damages lipids, proteins, and DNA, ultimately compromising the egg’s functionality. The balance between survival and death signals shifts as the expression and activation of apoptotic mediators such as Bax increase, while anti-apoptotic molecules like Bcl-xL decrease [291].

Translational control mechanisms also play a role. Upon aging, polyadenylation-induced translation of survival-associated mRNAs is downregulated [23]. Similarly, autophagy-related genes become dysregulated, and lysosomal activity fails to counterbalance cellular deterioration [295,296].

Taken together, these molecular events orchestrate the time-sensitive viability of unfertilized oocytes, serving as a quality control mechanism to eliminate non-functional eggs and prevent their accumulation in the reproductive system (Figure 1D).

### 7.4. Overactivation—Necrosis-like Cell Death of Unfertilized Eggs Due to Mechanical Stress

In addition to the well-characterized apoptotic and atretic pathways governing the fate of unfertilized oocytes, recent studies in *Xenopus laevis* and other model organisms have highlighted the phenomenon of overactivation, a distinct form of necrosis-like cell death that occurs in response to mechanical, osmotic, oxidative, or environmental perturbations [36,206,297,298,299,300,301]. This process is increasingly recognized as an underappreciated yet biologically relevant pathway through which post-mature, unfertilized eggs are eliminated.

Overactivation is characterized by a rapid influx of calcium, mitochondrial swelling, plasma membrane rupture, and the loss of ion homeostasis, distinct from the controlled dismantling seen in apoptosis [297]. Mechanical agitation or handling of eggs post-ovulation often triggers this response, especially in in vitro settings. Oxidative stress (e.g., hydrogen peroxide exposure) has been shown to induce overactivation in a time- and dose-dependent manner, accompanied by lipofuscin accumulation, loss of soluble cytoplasmic proteins, and intracellular ATP depletion [301] (Figure 1D).

Importantly, oxidative stress-induced overactivation is calcium-dependent and can be attenuated by the calcium chelator BAPTA. This pathway involves degradation of cyclin B2, decline of mitochondrial membrane potential (Δψm), ATP depletion, and termination of protein synthesis, all occurring within one hour of induction, but notably in the absence of caspase activation [299]. Similarly, spontaneous overactivation has been documented at low frequency in naturally spawned frog eggs. It is characterized by cortical contraction, egg swelling, ATP depletion, increased ADP/ATP ratio, cyclin B2 degradation, and plasma membrane rupture—hallmarks of necrotic cell death [298].

Key observations include: Loss of membrane integrity preceding nuclear changes. Mitochondrial dysfunction without the release of cytochrome c [297]. Absence of caspase activation, distinguishing it from canonical apoptosis [299]. Acute ATP depletion and cytoplasmic protein loss, marking necrotic collapse [298,301].

Overactivation may function as a fail-safe mechanism for clearing oocytes that are no longer viable or fertilizable. Recent findings suggest that mechanical stress during oviposition in the frog genital tract can trigger spontaneous overactivation, pointing to a physiological relevance beyond in vitro artifacts [298]. In the context of assisted reproduction or in vitro fertilization (IVF), minimizing physical and oxidative stress on oocytes is crucial to avoid triggering this destructive pathway. Moreover, this phenomenon represents an experimental variable when assessing egg viability and fertilization efficiency [302,303,304].

Further research is needed to: Identify molecular markers of overactivation in *Xenopus* and other species. Differentiate overactivation from apoptosis and necroptosis at the transcriptomic and proteomic levels. Explore thresholds of oxidative and mechanical stress that induce this pathway. Clarify the physiological roles of overactivation in natural reproductive contexts [36].

Understanding overactivation provides critical insight into oocyte physiology, laboratory best practices, and reproductive health, highlighting a dramatic contrast between the regulated activation of fertilized eggs and the catastrophic collapse of overactivated ones.

### 7.5. Clinical Implications—Intersection with Reproductive Medicine

The exploration of inactivated or degenerating oocytes in *Xenopus laevis* not only advances our understanding of fundamental developmental biology, but also offers translational insight into human reproductive medicine [94,305]. The post-maturation fate of oocytes—especially in the absence of fertilization—parallels key concerns in assisted reproductive technologies (ART), such as oocyte quality, timing of fertilization, and the efficacy of cryopreservation protocols [304,306].

Recent studies have highlighted the importance of cytoplasmic integrity, mitochondrial activity, and apoptotic resistance as benchmarks for oocyte viability in clinical settings [307,308]. The degeneration pathways—whether apoptotic, atretic, or necrotic—serve as reference points to evaluate and predict developmental competence. Furthermore, the deterioration of meiotic spindles and chromatin integrity, observed in *Xenopus* oocytes that fail to be fertilized, underscores the narrow temporal window for optimal fertilization in human oocytes [96].

In the realm of oocyte preservation, *Xenopus* models help refine cryoprotective agents and cooling protocols by offering real-time, visualizable readouts of membrane stability and organelle integrity [93,309,310]. Such insights also contribute to the development of diagnostic tools for assessing oocyte quality prior to in vitro fertilization (IVF), including molecular assays for oxidative stress markers or mitochondrial function.

As ARTs evolve, cross-species comparative studies with amphibian models like *Xenopus* remain critical. They allow clinicians and researchers to better anticipate cellular behaviors that affect fertility outcomes, and to design interventions that prolong oocyte viability or reverse premature activation [93,311].

### 7.6. Summary

Once *Xenopus* oocytes complete maturation, they enter a critical window during which fertilization must occur to initiate embryogenesis. In the absence of fertilization, matured eggs undergo a series of intrinsic molecular events that culminate in cellular degeneration. This chapter has examined the diverse fates of post-mature oocytes, revealing a spectrum of programmed and non-programmed death mechanisms that contribute to egg quality control and reproductive fidelity.

Three principal pathways define the demise of unfertilized eggs. First, apoptosis—characterized by caspase activation, cytochrome c release, and DNA fragmentation—represents a genetically encoded self-elimination program. Second, atresia encompasses the broader, heterogeneous degeneration of oocytes within the ovary, influenced by hormonal milieu and follicular interactions, often involving non-caspase-dependent processes. Third, recent studies highlight overactivation, a necrosis-like, calcium-dependent cell death triggered by oxidative stress, spontaneous metabolic imbalance, or mechanical stress during oviposition and in vitro handling. Overactivation is marked by rapid cortical contraction, ATP depletion, mitochondrial dysfunction, plasma membrane rupture, and the absence of caspase activity, distinguishing it from canonical apoptosis.

Collectively, these pathways underscore that oocyte death is not a passive decay but an actively regulated process ensuring the timely removal of defective or aged gametes. Understanding the molecular and temporal limits of egg viability has direct implications for assisted reproductive technologies (ART), including oocyte preservation, cryostorage optimization, and the development of biomarkers for egg quality.

By integrating classical knowledge with emerging insights on overactivation, *Xenopus* serves as a powerful model bridging basic developmental biology and translational reproductive medicine, offering lessons on gamete aging, infertility, and ovary preservation strategies.

## 8. Cutting-Edge Research Utilizing *Xenopus* Eggs (mRNA Injection Screening, Genome Editing)

The *Xenopus* egg has remained an invaluable model for developmental and molecular cell biology due to its large size, synchronous cell divisions, and ease of manipulation. Recent technological advances have further expanded its utility in cutting-edge research, particularly through mRNA injection-based functional screening and genome editing approaches.

One powerful application involves high-throughput mRNA injection screening. By systematically introducing synthetic mRNAs into fertilized eggs or oocytes, researchers can evaluate the function of genes in early developmental events such as axis formation, cell fate specification, and patterning [7,32,69,70,312,313]. For instance, overexpression or dominant-negative strategies using mRNA constructs encoding signaling molecules or transcription factors can reveal their roles in embryogenesis.

Moreover, the CRISPR/Cas9 system has revolutionized genome editing in *Xenopus*. Both *Xenopus laevis* and *Xenopus tropicalis* embryos can be injected with Cas9 protein or mRNA alongside guide RNAs (gRNAs) targeting genes of interest [314]. *X. tropicalis*, with its diploid genome, is particularly amenable to gene knockout studies. CRISPR technology enables precise mutagenesis, knock-in reporters, and targeted epigenome modifications. The availability of a reference genome sequence for *X. laevis* and *X. tropicalis* [6] has accelerated the design and interpretation of such genome-editing experiments.

Advances in imaging and transcriptomic tools also complement these manipulations. Live imaging of fluorescently tagged proteins and time-lapse microscopy allow visualization of dynamic developmental processes, while single-cell RNA-seq has provided detailed atlases of gene expression changes during early embryonic stages [32,67,68,315].

These integrative approaches enable *Xenopus*-based studies to tackle fundamental questions about morphogenesis, gene regulation, and cellular dynamics in a vertebrate system. At the same time, they offer experimental platforms for modeling human congenital diseases and testing therapeutic interventions.

## 9. Conclusions: Unresolved Questions and Future Directions

Despite decades of meticulous research using *Xenopus* oocytes and embryos, numerous questions remain unanswered, underscoring both the complexity and the richness of this model system. In this review, we have traced the journey of the frog egg—from the molecular architecture of the oocyte, through its meiotic maturation and fertilization, to either the orchestration of early developmental programs or its demise when fertilization fails. Each step reveals not only a finely tuned regulatory system but also critical knowledge gaps that continue to intrigue developmental biologists.

One persistent challenge is the full elucidation of the spatial–temporal regulation of signaling pathways within the large cytoplasm of the oocyte. While we have gained insights into MPF activation, MAPK cascades, and translational control elements like CPEB and polyadenylation dynamics, the interplay among these elements in three-dimensional space and over time remains incompletely understood. Emerging imaging technologies, such as high-resolution light-sheet microscopy and in vivo biosensors, offer promising tools to address these complexities.

The mechanisms underlying the decision between survival and programmed cell death in post-ovulatory, unfertilized eggs represent another area of active inquiry. Whether through apoptotic, necrotic, or novel hybrid pathways, the fate of the mature egg cell is intimately tied to its metabolic and mitochondrial status. Clarifying these mechanisms could have far-reaching implications for reproductive biology, fertility preservation, and the development of egg quality diagnostics.

Furthermore, while *Xenopus laevis* has provided a powerful platform for discovery, the advent of *Xenopus tropicalis*—with its diploid genome and amenability to genetic manipulation—opens new frontiers in forward genetics and evolutionary developmental biology (evo-devo). Cross-species comparisons can illuminate conserved and divergent mechanisms of oocyte biology and early development.

Future research will also benefit from interdisciplinary approaches that integrate transcriptomics, proteomics, and systems biology, potentially revealing emergent properties of developmental systems. Additionally, incorporating AI-driven image analysis and predictive modeling may help generate new hypotheses and deepen our understanding of cell fate decisions in development.

In sum, the *Xenopus* oocyte continues to be not only a subject of fascination but also a launching point for addressing fundamental questions about life’s beginnings. As we refine tools and expand conceptual frameworks, this enduring model organism will remain central to decoding the molecular logic of development.

## Figures and Tables

**Figure 1 biomolecules-16-00022-f001:**
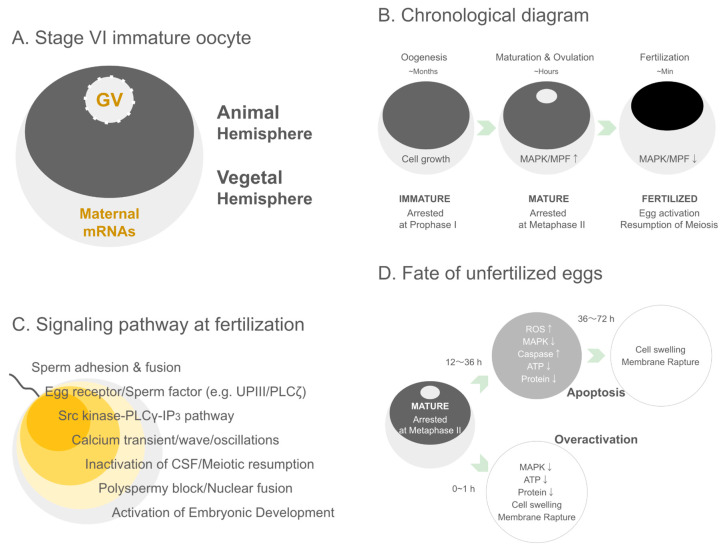
Structural, developmental, signaling, and post-ovulatory features of *Xenopus* oocytes and eggs. (**A**) Stage VI immature oocyte. A schematic representation of a fully grown *Xenopus* stage VI oocyte showing the characteristic animal–vegetal polarity. The darkly pigmented animal hemisphere contains the germinal vesicle (GV), while the vegetal hemisphere is enriched in maternal mRNAs that will direct early embryonic patterning after fertilization. (**B**) Chronological diagram. A timeline illustrating the major transitions of *Xenopus* oocytes from prolonged oogenesis to meiotic maturation and fertilization. During oogenesis (months), the oocyte grows while remaining arrested in prophase I. Upon hormonal stimulation, meiotic maturation and ovulation occur over several hours, accompanied by increased MAPK and MPF activity and arrest at metaphase II. Following fertilization (minutes), MAPK and MPF activities rapidly decline, leading to egg activation, release from metaphase arrest, and the resumption of meiosis. (**C**) Signaling pathway at fertilization. A schematic summary of the major signaling events triggered upon fertilization in *Xenopus*. Following sperm adhesion and fusion, egg surface receptors and sperm-derived factors (e.g., uroplakin III and PLCζ) initiate signaling cascades. Activation of the Src kinase–PLCγ–IP_3_ pathway induces a characteristic Ca^2+^ transient/wave/oscillatory response, leading to CSF inactivation and meiotic resumption. These Ca^2+^-dependent events also promote the block to polyspermy, subsequent pronuclear fusion, and ultimately the initiation of embryonic development. (**D**) Fate of unfertilized eggs. A schematic representation of the two major post-ovulatory fates of unfertilized *Xenopus* eggs. Mature eggs arrested at metaphase II undergo time-dependent deterioration. After 12–36 h, gradual declines in MAPK activity, ATP levels, and protein content—together with increases in ROS production and caspase activation—lead to apoptosis, culminating in cell swelling and membrane rupture by 36–72 h. Alternatively, within 0–1 h after ovulation, eggs may undergo overactivation, characterized by premature loss of MAPK activity, ATP depletion, rapid cytoplasmic deterioration, and early membrane rupture. These distinct pathways underscore the metabolic and signaling vulnerabilities of unfertilized post-ovulatory eggs.

## Data Availability

Not applicable.

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
