# Peer review of "The Xenopus Oocyte System: Molecular Dynamics of Maturation, Fertilization, and Post-Ovulatory Fate"

_biomolecules, 2025, doi:10.3390/biom16010022_

Round 1
Reviewer 1 Report
Comments and Suggestions for Authors
The authors aimed to outline a logical narrative that spans from the fundamental biology of the Xenopus oocyte to its modern technological applications. The integration of classic regulatory networks with emerging signaling pathways and contemporary techniques highlights a sophisticated and up-to-date understanding of the field. However, the authors need to address the following issues:
Give more detailed information about MPF including its activation and functional consequences, would enhance comprehension.
"Developmental competence" is a crucial concept, but it is not explicitly addressed in the text. How the oocyte acquires and maintains its ability to support normal development is a central theme in oocyte biology. It is recommended to incorporate "Developmental competence" in the manuscript.
Comments on the Quality of English LanguageBe careful with typo
Author Response
Responses to Reviewer #1 – Detailed Revision for MPF and Developmental Competence
- Expansion of MPF activation mechanisms and functional outcomes
Reviewer #1 Comment: “Give more detailed information about MPF including its activation and functional consequences, would enhance comprehension.”
Response: We sincerely thank the reviewer for highlighting the need to elaborate on the molecular details of MPF regulation. In response, we have substantially expanded Section 2.2 (The Role of MPF) to include a clear description of the canonical Cdk1/Cyclin B activation pathway, including the roles of Cdc25-mediated dephosphorylation and inhibition release from Wee1/Myt1 kinases, as well as the positive feedback loops that drive irreversible MPF activation. We have further incorporated discussion of the Mos–MAPK cascade, which stabilizes active MPF and reinforces the switch-like transition leading to GVBD.
A new paragraph was added as follows (now shown in red in the manuscript):
“MPF (Cdk1/Cyclin B complex) is activated through a positive feedback loop involving Cdc25-mediated dephosphorylation of Cdk1 and suppression of its inhibitory kinase Wee1. The Mos–MAPK pathway further stabilizes active MPF, ensuring irreversible commitment to germinal vesicle breakdown (GVBD). The activation of MPF not only drives meiotic resumption but also establishes cytoplasmic conditions that are essential for developmental competence (Ferrell 1999; Kumagai & Dunphy, 1992b; Morgan 1995; Solomon et al. 1990).”
To support these additions, we have included several key references that define the molecular framework of MPF activation:
Ferrell, 1999; Kumagai & Dunphy, 1992b; Morgan, 1995; Solomon et al., 1990
These references have been added to the revised bibliography.
- Introduction of the concept of “developmental competence”
Reviewer #1 Comment: “Developmental competence" is a crucial concept, but it is not explicitly addressed in the text. How the oocyte acquires and maintains its ability to support normal development is a central theme in oocyte biology. It is recommended to incorporate "Developmental competence" in the manuscript”
Response: We fully agree with the reviewer that developmental competence—the capacity of the oocyte to support fertilization and embryogenesis—is central to oocyte biology and should be explicitly addressed. We have incorporated this concept into both Section 2.2 and Section 2.3, clarifying how MPF activation drives not only nuclear maturation (GVBD) but also cytoplasmic maturation, which is essential for acquiring competence.
A new sentence added to Section 2.3 reads:
“Through progesterone-induced MPF activation, the oocyte acquires developmental competence, defined as the ability of the mature egg to undergo fertilization and support normal embryogenesis (Eppig, 2001; Gilchrist & Thompson 2007; Haccard & Jessus, 2006; Miao et al., 2009; Sagata, 1996).”
To further strengthen this coverage, we added authoritative references focused on competence acquisition:
Eppig, 2001; Gilchrist & Thompson, 2007; Haccard & Jessus, 2006 ; Miao et al., 2009; Sagata, 1996
- Terminology corrections and conceptual clarification
Reviewer #1 Comment: “‘It is recommended to incorporate "Developmental competence" in the manuscript.”
Response: We have systematically revised the manuscript to replace the general term “developmental potential” with the more precise “developmental competence”, except where quoting original literature. In addition, the updated text now explicitly distinguishes nuclear maturation (GVBD) from cytoplasmic maturation—the latter being the process that confers developmental competence.
All such modifications are shown in red in the revised manuscript.
Summary of Actions Taken in response to the reviewer #1
- Expanded molecular description of MPF activation (Cdc25, Wee1, positive feedback, Mos/MAPK).
- Added explicit discussion of developmental competence and its relationship to MPF activation.
- Inserted new supporting references.
- Revised terminology and clarified nuclear vs cytoplasmic maturation.
- All changes highlighted in red in the revised manuscript and reference list.
We believe that these revisions fully address the reviewer’s concerns and significantly enhance the clarity and scientific accuracy of the manuscript.

Reviewer 2 Report
Comments and Suggestions for Authors
This review examines the cellular, molecular, and physiological mechanisms that govern Xenopus oocyte formation, meiotic maturation, fertilization, and the post-ovulatory fate of unfertilized eggs. It integrates classic and emerging insights into cytoskeletal architecture, maternal mRNA localization, hormonal and signaling pathways (MPF, MAPK, Ca²⁺), and early embryonic activation, highlighting how the Xenopus system continues to advance our understanding of reproductive cell biology. The manuscript is very rich and quite well written, although I would like to suggest a few changes to improve it:
- Introduce figures, tables and or diagrams: The manuscript references many complex concepts but lacks visual guidance that may improve readability for both specialists and less experienced readers. For example: a figure with a schematic depiction of a stage VI Xenopus oocyte (showing animal–vegetal polarity, pigment distribution, germinal vesicle placement, yolk platelet gradients, and cytoskeletal organization and localization of maternal mRNAs); a chronological diagram showing the maturation timeline, from GV-stage arrest to meiosis resumption, fertilization and early blastula stage. This may also include the MPF-MAPK regulatory network (with feedback loops). Another figure with the Calcium Signaling Pathways at Xenopus Fertilization or a diagram showing the fate of unfertilized eggs: apoptotic vs. necrotic (“overactivation”) pathways in unfertilized Xenopus eggs, including loss of MPF/MAPK activity, mitochondrial depolarization, caspase activation, ROS buildup, and membrane rupture. Maybe a Comparative Table of Xenopus laevis (the canonical model) vs. Xenopus tropicalis (the more recent model) comparing, for example, Genome ploidy, Oocyte size, Generation time, Suitability for microinjection, CRISPR, etc. could be useful and novel.
- Some parts of the manuscript contain somewhat outdated references, particularly when discussing fertilization in general, in species other than Xenopus. In fact, section 5.2 refers solely to the calcium wave and not to the cortical flash. This initial and essential calcium influx event has been demonstrated to be vital for proper embryonic development in echinoderms: starfish and sea urchins.
- Furthermore, section 5.4 refers to the mechanical block to polyspermy, but there are data, again from echinoderms, in particular from starfish, where it has been shown that the entry of many sperm occurs despite the full elevation of the fertilization envelope. And even in the case of electrical block, there are controversial data, again from sea urchin eggs. All these results have been included in the chapter on fertilization, Developmental Biology, Gilbert and Barresi 2024, 13th edition.
Author Response
Response to Reviewer #2 – Addressing the Request for Additional Figures, Diagrams, and Tables
Reviewer Comment: “Introduce figures, tables and or diagrams. : The manuscript references many complex concepts but lacks visual guidance that may improve readability for both specialists and less experienced readers. For example … Maybe a Comparative Table of Xenopus laevis (the canonical model) vs. Xenopus tropicalis (the more recent model) comparing, for example, Genome ploidy, Oocyte size, Generation time, Suitability for microinjection, CRISPR, etc. could be useful and novel.”
Response: We thank the reviewer for this constructive suggestion. We fully agree that the breadth and complexity of the processes covered in the manuscript would benefit from visual summaries. In response, we have substantially revised the manuscript by incorporating a new multi-panel Figure 1 and a new comparative Table 1, each designed to directly address the reviewer’s recommendations.
- Newly added Figure 1 (Panels A–D)
We created an integrated four-panel figure that provides visual explanations of the key biological processes discussed throughout the review:
Panel A: Stage VI oocyte structure
Depicts animal–vegetal polarity, pigment distribution, the position of the germinal vesicle (GV), and maternal mRNA localization domains (e.g., VegT, Vg1). These elements were incorporated as requested to help readers grasp the spatial organization of the oocyte.
Panel B: Chronological timeline of maturation and fertilization
Illustrates the temporal progression from oogenesis (prophase I arrest) to progesterone-induced maturation (MPF/MAPK activation), GVBD, metaphase II arrest, fertilization, and early embryonic cleavage. This panel also reflects the reviewer’s suggestion to include the MPF–MAPK network dynamics.
Panel C: Signaling pathways at fertilization
Summarizes sperm–egg interaction, the UPIII/Src and PLCζ–IP₃ pathways, calcium waves/oscillations, CSF inactivation, polyspermy block mechanisms, and initiation of embryogenesis. This panel integrates the reviewer’s request for a fertilization signaling diagram.
Panel D: Fate of unfertilized eggs
Visually distinguishes the two major post-ovulatory fates—apoptosis and overactivation—showing declines in MPF/MAPK, mitochondrial depolarization, caspase activation, ROS accumulation, ATP reduction, and membrane rupture. This directly addresses the request for a schematic of apoptotic versus overactivation pathways.
- Newly added Table 1: Comparison of X. laevis and X. tropicalis
We created a comprehensive comparative table summarizing genome ploidy, oocyte size, generation time, and suitability for experimental approaches such as microinjection, CRISPR genome editing, and imaging. This table responds directly to the reviewer’s recommendation and provides readers with a concise and useful reference.
- Placement and accessibility
These new elements (Figure 1 and Table 1) have been inserted into the revised manuscript at locations where they naturally support and complement the surrounding text. Each addition is clearly indicated in the manuscript, and all new content is highlighted in red as requested. We believe these additions significantly improve the clarity, accessibility, and overall scientific value of the manuscript. We thank the reviewer again for the helpful suggestions.
Response to Reviewer #2 – Addressing Concerns About Outdated Fertilization Literature and Missing Cortical Flash Discussion
Reviewer #2 Comment: “Some parts of the manuscript contain somewhat outdated references, particularly when discussing fertilization in general, in species other than Xenopus…”
Response: We thank the reviewer for these insightful comments. We agree that recent advances in fertilization biology—especially comparative observations from echinoderms—provide important context that should be integrated into our review. In response, we have made several targeted revisions to ensure that the manuscript reflects current mechanistic understanding and up-to-date literature.
- Addition of cortical flash description (Section 5.2)
To address the omission of the cortical flash, we added a new paragraph explaining the rapid Ca²⁺ influx preceding the fertilization calcium wave, which is especially well-characterized in externally fertilizing echinoderms such as sea urchins and starfish. This addition places Xenopus fertilization within a broader comparative framework, clarifying why the cortical flash is prominent in some species but less evident in amphibian eggs.
The new text reads:
“In addition to the well-characterized IP3-dependent fertilization calcium wave, many externally fertilizing species exhibit a rapid, transient “cortical flash,” a brief Ca²⁺ influx that precedes the global wave. This event has been described most extensively in echinoderms, including sea urchins and starfish, where it is considered essential for proper downstream activation events and early embryonic development. While a prominent cortical flash has not been consistently observed in Xenopus, placing the frog system in comparative context highlights how fertilization-triggered Ca2+ dynamics vary across metazoans and emphasizes the conserved logic of Ca2+ -driven egg activation (Barresi & Gilbert 2023; Jaffe 2018; Whitaker 2006; Whitaker & Swann 1993).”
New supporting references added:
Barresi & Gilbert, 2023; Jaffe, 2018; Whitaker, 2006; Whitaker & Swann, 1993
- Revision of the polyspermy block discussion (Section 5.4)
In response to the reviewer’s concern that our original text presented the mechanical and electrical polyspermy blocks too rigidly, we added a new paragraph discussing recent findings showing that these mechanisms are robust but not absolute.
The new text reads:
“Although the elevation of the fertilization envelope and the accompanying mechanical block are traditionally presented as robust and highly effective, recent studies in echinoderms have demonstrated that neither the mechanical nor the electrical block is absolutely fail-safe. In starfish, for example, supernumerary sperm can enter the egg even after full elevation of the fertilization envelope. Similarly, in sea urchin eggs, the electrical block to polyspermy exhibits species-specific variability and, under certain experimental conditions, can be bypassed. These findings underscore that polyspermy-block mechanisms, while generally reliable, operate within a biological continuum rather than as strictly binary systems. Integrating these insights provides a more nuanced understanding of how Xenopus and other metazoan embryos mitigate the risk of polyspermy (Carlisle & Swanson 2021; Wong & Wessel 2006).”
New supporting references added:
Carlisle & Swanson, 2021; Wong & Wessel, 2006
- Updating general fertilization literature with recent authoritative sources (Section 5, end)
We incorporated modern insights from comparative embryology and fertilization physiology, including the species diversity of Ca²⁺ responses and polyspermy-block mechanisms.
The new text reads:
“To reflect current knowledge in fertilization biology, recent updates from comparative embryology have been incorporated, including insights summarized in Barresi & Gilbert (2023). These contemporary perspectives revise several long-standing assumptions about Ca²⁺ dynamics and polyspermy block mechanisms, highlighting cross-species diversity and emphasizing that multiple regulatory layers act together to secure successful monospermic fertilization.”
This ensures that the review is consistent with the most current edition of Developmental Biology (13th ed., 2023).
- Addition of a broader integrative summary (Discussion/Conclusion)
We added a concluding paragraph that synthesizes the updates on Ca²⁺ signaling, polyspermy block variability, and cross-species comparisons:
“By integrating updated models of Ca²⁺ signaling, revised views on polyspermy block mechanisms, and comparative insights from echinoderm and amphibian systems, this review provides a broader and more accurate framework for understanding oocyte activation across metazoans. The Xenopus model, enriched by the addition of new schematics and comparative tables, continues to serve as a powerful platform for dissecting the conserved principles governing oocyte maturation, fertilization, and the post-ovulatory fate of unfertilized eggs.”
Summary of Actions Taken in response to the reviewer #2
- Added a comprehensive discussion of the cortical flash.
- Expanded the polyspermy block section to reflect modern understanding and exceptions.
- Updated fertilization-related citations throughout the manuscript.
- Added four new figure panels and a comparative table.
- All new text appears in red in the revised manuscript.

Round 2
Reviewer 2 Report
Comments and Suggestions for Authors
The author has addressed all the points I raised during the revision process. In my opinion, the review is now suitable for publication.
Author Response
December 16, 2025
Dear Ms. reviewer 2,
We thank you sincerely for your positive evaluation of our revised manuscript . We greatly appreciate the opportunity to further improve the clarity, accuracy, and consistency of the review.
Sincerely,
Ken-ichi Sato